# Implication of the Sensory Environment in Children with Autism Spectrum Disorder: Perspectives from School

**DOI:** 10.3390/ijerph18147670

**Published:** 2021-07-19

**Authors:** Ana Gentil-Gutiérrez, José Luis Cuesta-Gómez, Paula Rodríguez-Fernández, Jerónimo Javier González-Bernal

**Affiliations:** 1Department of Health Sciences, University of Burgos, 09001 Burgos, Spain; ana.gentil@autismoburgos.org (A.G.-G.); jejavier@ubu.es (J.J.G.-B.); 2Department of Educational Sciences, University of Burgos, 09001 Burgos, Spain

**Keywords:** autism, ASD, children, sensory difficulties, processing patterns, sensory systems, school

## Abstract

(1) Background: Children with Autism Spectrum Disorder (ASD) frequently have difficulties in processing sensory information, which is a limitation when participating in different contexts, such as school. The objective of the present study was to compare the sensory processing characteristics of children with ASD in the natural context of school through the perception of professionals in the field of education, in comparison with neurodevelopmental children (2) Methods: A cross-sectional descriptive study as conducted with study population consisting of children between three and ten years old, 36 of whom were diagnosed with ASD and attended the Autismo Burgos association; the remaining 24 had neurotypical development. The degree of response of the children to sensory stimuli at school was evaluated using the Sensory Profile-2 (SP-2) questionnaire in its school version, answered by the teachers. (3) Results: Statistically significant differences were found in sensory processing patterns (*p* = 0.001), in sensory systems (*p* = 0.001) and in school factors (*p* = 0.001). Children with ASD who obtained worse results. (4) Conclusions: Children with ASD are prone to present sensory alterations in different contexts, giving nonadapted behavioral and learning responses.

## 1. Introduction

The neologism autism derives from the Greek prefix autós, which means “shelf”. Eugen Bleuler first used this term in 1911 as a characteristic of schizophrenia, but it was not until the 1940s that it was formally recognized as a syndrome following the description made by Kanner and Asperger [1], who identified difficulties in communication and social interaction in patients with this type of pathology [2].

According to the fifth edition of the Diagnostic and Statistical Manual of Mental Disorders (DSM-V), autism is classified within neurodevelopmental disorders. It is characterized by difficulties at the social level, both in interaction and in communication, and by the presence of restricted and repetitive patterns at the behavioral level. It is a specific personal continuum that begins in childhood and is related to all areas of development. At the sensory level, possible difficulties are evident in repetitive and restricted patterns of behaviors, activities and interests [3].

The population with Autism Spectrum Disorder (ASD) is susceptible to sensory alterations. Studies in which this circumstance is manifested are described in the literature [4,5]. Kanner [1] described how children diagnosed with ASD reacted pleasantly or negatively towards certain sensory stimuli, and Jean Ayres [6] documented for the first time the differences between children in sensory processing. Sensory processing is a broad term that usually includes the direction of sensory information by the neural systems, including the central and peripheral nervous systems, and the functions of the receptor organs. Sensory processing dysfunction can present in the form of hyposensitivity or hypersensitivity to stimuli [4,5], which often elicits behavioral responses that are unusual or different from those expected in a neurotypical person [7,8,9,10].

Up to 90% of people with ASD experience some form of sensory hypersensitivity, which has been shown to be one of the main causes of disruptive behavior [11]. These behaviors are aggravated by limitations to manage self-regulation presented by this group, as well as by difficulty in identifying and describing feelings and accurately self-reporting on physiological sensations or mood [12]. Disruptive and problematic behaviors acquire even more relevance in the cases of children with ASD with associated intellectual disabilities and/or impaired verbal abilities [12].

A growing body of research suggests that atypical sensory processing may be a central phenotype in autism due to its link with higher-order cognitive and social symptoms, and its potential to serve as an early diagnostic marker [13]. In addition, alterations in sensory processing have significant impacts on the development of the child with autism, limiting their participation in socialization contexts, such as school [14], and making it difficult to function in areas of occupation, such as activities of daily living (ADL) [15,16,17,18], play [19,20] or education [21,22].

Taking into account the above, it is essential to study the perception of adults in different environments, so that the problem may be addressed in the largest number of contexts. Therefore, this research constitutes a starting point for the development of new hypotheses, the main objective being to compare the characteristics of sensory processing of children with autism in the natural context of school through the perception of professionals in the field of education, in comparison with neurodevelopmental children.

## 2. Materials and Methods

### 2.1. Study Design—Study Population

A descriptive cross-sectional study was undertaken with a population consisting of children between three and 10 years old. The sample was divided into two groups: on the one hand, there were children with normal or neurotypical neurological development enrolled in a public school in Burgos; on the other hand, the second group was made up of children with a diagnosis of ASD enrolled in the same public school and who went to the Autismo Burgos facilities to receive outpatient support services. Autismo Burgos is a nonprofit association promoted by relatives of people with ASD whose purpose is to improve quality of life, providing specific and specialized support throughout the life cycle and in all areas of the attendee’s life. It was declared a Public Utility Entity in 1999 and has been certified in Quality according to the ISO 9001 Standard since 2002 in all its centers and services [23].

Children with severe disruptive behaviors who suffered from another type of diagnosed serious mental disorder were excluded from the study.

### 2.2. Procedures

Participants were recruited through a convenience sample, being the professionals from Autismo Burgos who selected the children with ASD and the participants from the neurotypical group who met the inclusion criteria. Prior to the start of the study, a meeting was held with the children’s parents and teachers in which the purpose of the study and its voluntary nature were explained to them. If they agreed to participate, they were asked to sign the informed consent. Education professionals, both of children with ASD and neurotypical children, were provided with the information collected through a self-administered online questionnaire through Google Drive (Google LLC., Mountain View, CA, USA) in March 2019 for subsequent statistical analysis.

The Bioethics Committee of the University of Burgos approved the research, (Reference IR 14/2019), respecting at all times the requirements established in the Declaration of Helsinki of 1975.

### 2.3. Main Outcomes—Instruments

The main outcome variable of the study was the degree of response of the children to sensory stimuli at school. As an evaluation tool, the Sensory Profile-2 (SP-2) questionnaire was used in its school version, based on Dunn´s sensory processing model, which allowed us to analyze how the processing interfered with the child´s participation in different environments [24]. This is an instrument validated in the Spanish population for children between the ages of three and 14 years, made up of 44 items, which assesses the frequency with which the child shows certain behaviors in familiar situations and environments through a standard Likert scale of six possible response options (5 = almost always or always, 4 = frequently, 3 = half the time, 2 = occasionally, 1 = almost never or never, and 0 = not applicable). The items were classified into patterns of sensory processing or quadrants (search engine, as degree of obtaining sensory information; avoidant, to what extent sensory inputs can be annoying; sensitive, how sensory inputs are detected; spectator, level with which they are unaware or ignore sensory inputs), sensory and behavioral systems or sections (auditory, visual, tactile, movement, behavioral), and school factors (student need for external support to participate in learning activities, awareness and attention for learning, tolerance of the student in the school environment, and the availability of the student to learn in the educational context) [24,25]. The evaluation instrument has good psychometric properties, with a Cronbach’s alpha that ranges between 0.72 and 0.90 and a test stability coefficient between 0.87 and 0.97 [24].

Variables of interest such as age and gender, were also collected to obtain relevant sociodemographic information.

### 2.4. Statistical Analysis

A descriptive analysis was carried out to express the sociodemographic characteristics of the sample and the scores obtained in the primary outcome variable of the study. Categorical variables were expressed as absolute frequencies and percentages, while continuous variables were expressed as means. Compliance with the normality criteria of the quantitative variables was evaluated using the Kolmogorov-Smirnov test. To evaluate the association between the means reported by the education professionals in the different sections of the SP-2 questionnaire, and the type of group to which the evaluated children belonged (ASD or neurotypical), the nonparametric Mann-Whitney U test was performed.

Statistical analysis was performed with SPSS version 25 software (IBM-Inc, Chicago, IL, USA). For the analysis of statistical significance, a value of *p* < 0.05 was established.

## 3. Results

The study sample consisted of 60 children, aged between three and 10 years old at the time of completing the sensory profiles. Of the total number of participants, 36 were part of the group of children with ASD, 29 being male (80.6%) and seven female (19.4%). Twenty-four participants made up the group of neurotypical children. Gender was equally distributed, with 12 girls and 12 boys.

Table 1 summarizes the mean scores obtained in the category “sensory processing patterns” of the SP-2 questionnaire by the children of the ASD group and the neurotypical group in the school context. Statistically significant differences were found between the scores reported by educational professionals who worked with children with ASD and those provided by teachers of neurotypical children. Students with ASD showed significant alterations in all sensory processing patterns compared to neurotypical students.

Regarding the sensory and behavioral systems, educational professionals who worked with children with ASD reported statistically significant dysfunction in this group, compared to neurotypical children, after evaluating their auditory, visual, tactile and movement processing. Differences were also found in the behavioral systems, with the teachers of the children with ASD being those who provided data on behavioral aspects that were significantly more dysfunctional than the teachers of the neurotypical children (Table 2).

Considering school factors, the teachers of children with ASD reported a greater need for external help, less awareness and attention to learning, less tolerance in the learning context and less availability to learn in this group. Statistically significant differences were found in the four factors after comparing the scores obtained in the children with ASD with the group of neurotypical children (Table 3).

## 4. Discussion

The study of sensory behaviors in education is important because school is a highly stimulating context in which children have less control of the environment and increased learning demands [26]. Previous studies have shown a higher prevalence of hypersensitivity in children with ASD compared to neurotypical children, which can be exacerbated in contexts such as school, contributing to maladaptive behavioral and learning responses [27]. The present research is based on this aspect, which showed an alteration in the sensory processing of children with ASD in the educational context compared to children with neurotypical development, as well as a greater need for support.

Our study showed a significant alteration in the four patterns of sensory processing, according to Dunn [18], in children with ASD at school. Mills et al. [28] reported in their study that students with ASD may experience difficulties in performing classroom tasks due to atypical sensory processing and inefficient use of higher-order cognitive strategies, because they frequently respond by avoiding certain situations, seeking sensory experiences, experiencing fear or even acquiring a passive role [29,30,31,32]. Along the same lines, a recent study demonstrated several sensory integration disorders in a group of children with ASD at school that included problems in sensory modulation, sensory discrimination and perceptual dysfunctions, vestibular difficulties, dyspraxia, and sensory-seeking behaviors [33]. Children with hypersensitivity may respond in the form of fear, avoidance, distraction, excessive vigilance or even aggression towards specific sensations [6,26], hindering adaptive behavior and preventing the performance of daily activities, academic skills and social participation [34].

In addition to promoting maladaptive responses, sensory processing abnormalities in children with ASD can appear in different sensory modalities [34]. Previous studies have shown atypical sensory disturbances in relation to inference to pain, avoidance of certain sounds or textures, aversion to unusual smells or unknown objects, or the search for specific visual experiences [35,36]. These results are confirmed with those found in the present investigation, where children with ASD showed greater dysfunction in all sensory and behavioral systems compared to neurotypical children. Likewise, a study carried out by Fernández et al. [37] to compare sensory processing in children with and without autism spectrum disorder at home and in the classroom found that the most affected sensory modalities in the group of children with ASD were hearing and touch. A tactile or auditory defensive attitude were the most frequent manifestations of children with hypersensitivity, which is known to be one of the most common responses of children with ASD [38].

Recent research showed lower participation in school in children with sensory disorders [39]. Taking into account the different school factors studied in our research, teachers perceived a greater need for external help, less awareness and attention to learning, less tolerance in the learning context and less availability to learn, in students with ASD. This aspect can be explained by the fact that children with difficulties in the treatment and processing of sensory information develop less interest in school activities, as well as maladaptive behaviors such as avoidance, which can affect their academic results and their performance in the classroom. [39]. The elevated response to sensory input due to the increasing demands of the environment, coupled with the child´s limited control over the particular educational environment, may negatively influence the way they behave in school [27].

It has been proposed that the ability to perceive internal or external information in a modulated way allows focusing and maintaining attention, approaching and exploring the environment with confidence, and responding to the demands of the environment in an adaptive way [40]. Sensory and social behaviors can exert a reciprocal influence on each other, where atypical sensory processing can affect selective attention to different stimuli, social reciprocity, or even adherence to social norms of behavior [41]. Sensory-seeking behavior is prevalent among children with ASD, which could be also used to help dampen or cancel over reactivity to certain stimuli, avoiding behaviors that are not socially appropriate or safe [32].

This research provides information about the involvement of the sensory environment in school in a sample of children with ASD compared to a group of children with neurotypical development but must be considered in the context of its limitations. The use of a self-administered online survey and nonprobability sampling can lead to methodological biases. Furthermore, the sample size was not representative for the general population, and the fact that it was a cross-sectional study made it impossible to determine a causal relationship between the variables. Despite this, having a group of children with neurotypical development facilitated the identification of atypical sensory development in the participants with ASD.

Although there are studies that describe the sensory profiles of children with ASD and neurotypical children, it is essential to continue with the research so that families and professionals understand their sensory modulation processes before considering objectives and intervention methods focused on improving functional abilities, providing adequate spaces based on their characteristics. In general, children with ASD require sensory adjustments in the school environment that reduce the tendency to hypersensitivity and facilitate their self-regulation, but each child has different sensory processing. These data related to the sensory profile of children with ASD at school provide information about the child´s strengths and challenges at the sensory level in this context, help to distinguish factors that vary in their behavior based on their sensory experiences, and facilitate adapting the school environment to personal needs.

Future research is recommended to evaluate the sensory profile of children with and without different pathologies and to carry out interventions in different natural environments, to enhance the ability of children to cope effectively with different stimuli of the school context. Despite the sensory peculiarities of each child, it is possible to extract some practical implications at a general level, not so much to implement and design specific interventions, but to adapt the environment to the needs of each child considering deficiencies in aspects such as auditory and tactile processing, or eliminating possible threatening stimuli in the classroom. In addition, developing guidelines for parents and teachers of children with atypical sensory profiles can contribute to a better occupational performance of the child and promote appropriate social behaviors. A change of perspective is considered essential, so that it is the school establishment that understands and adapts to the needs of children with ASD.

## 5. Conclusions

The different forms of sensory processing of children with ASD, compared to the neurotypical children, may be related to their development and influence the performance of occupations in different contexts, such as school. After comparing the sensory profile of children with ASD with neurotypical children, statistically significant differences were found in sensory processing patterns, sensory systems and school factors, with children with ASD having the worst results. These data facilitate adapting the school environment to personal needs, since a change of perspective is considered essential in which it is the school establishment that understands and adapts to the needs of children with ASD.

## Figures and Tables

**Table 1 ijerph-18-07670-t001:** Comparison of the scores obtained by children with ASD and neurotypical children in the category “sensory processing patterns”.

Sensory Processing Patterns	Group	*n*	Rank	U	*p*-Value
Search engine	ASD	36	42.43	2.50	0.001
	Neurotypical	24	12.60		
Avoidant	ASD	36	42.12	13.50	0.001
	Neurotypical	24	13.06		
Sensitive	ASD	36	40.76	62.50	0.001
	Neurotypical	24	15.10		
Spectator	ASD	36	41.65	30.50	0.001
	Neurotypical	24	13.77		

*n*: Sample size.

**Table 2 ijerph-18-07670-t002:** Comparison of the scores obtained by children with ASD and neurotypical children in the category “sensory and behavioral systems”.

Sensory and Behavioral Systems	Group	*n*	Rank	U	*p*-Value
Auditory	ASD	36	39.67	102.00	0.001
	Neurotypical	24	16.75		
Visual	ASD	36	42.36	5.00	0.001
	Neurotypical	24	12.71		
Tactile	ASD	36	38.44	146.00	0.001
	Neurotypical	24	18.58		
Movement	ASD	36	41.33	42.00	0.001
	Neurotypical	24	14.25		
Behavioral	ASD	36	42.36	5.00	0.001
	Neurotypical	24	12.71		

*n*: Sample size.

**Table 3 ijerph-18-07670-t003:** Comparison of the scores obtained by children with ASD and neurotypical children in the category “school factors”.

School Factors	Group	*n*	Rank	U	*p*-Value
External support	ASD	36	42.17	12.00	0.001
	Neurotypical	24	13.00		
Awareness and attention for learning	ASD	36	42.19	11.00	0.001
	Neurotypical	24	12.96		
Tolerance	ASD	36	42.31	7.00	0.001
	Neurotypical	24	12.79		
Availability to learn	ASD	36	40.07	87.50	0.001
	Neurotypical	24	16.15		

*n*: Sample size.

## Data Availability

All of the data are available in the manuscript.

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
