# Peer review of "Implication of the Sensory Environment in Children with Autism Spectrum Disorder: Perspectives from School"

_ijerph, 2021, doi:10.3390/ijerph18147670_

Round 1

Reviewer 1 Report

The current manuscript entitled as ‘Implication of the sensory environment in children with Autism Spectrum Disorder: perspectives from school’ by Ana Gentil-Gutiérrez et al. is well written. The pattern of this manuscript is appreciable. No further actions are required. The authors are requested to check grammatical errors throughout the manuscript. In addition, they are suggested to provide the pattern of questionnaire as a supplemental information so that future researchers can manage it easily. 

Reviewer 2 Report

Revision of the article:

Implication of the sensory environment in children with Autism Spectrum Disorder: perspectives from school

This article compares the sensory processing characteristics of children with Autism Spectrum Disorder (ASD) with neurodevelopmental children in the context of school.

The introduction is very clear and the authors perfectly justify the objective of this research. The methodologically design and results are well explained. As limitations, note the small sample size, that the sample was at convenience and the cross-sectional nature of the study, but the authors already mention it in the limitations section.

However, before evaluating its publication, it is necessary to correct the References section according to the journal's instructions. This request is made according to the reference standards of the journal: https://www.mdpi.com/journal/ijerph/instructions

Kind regards

Reviewer 3 Report

The theoretical framework should be expanded to provide solid support from previous research on the subject. On the other hand, it should include more up-to-date references on the subject.

The discussion should debate the results obtained in greater depth, including a larger and more up-to-date bibliography.

It is necessary to offer a more powerful and well-argued academic discourse.

It is not clear and the novelty of the study and its findings should be emphasised throughout the work and fundamentally in the conclusions.

The practical implications and the implementation of the results in the reality of education should be improved.

At the methodological level, in general, there are significant shortcomings. The authors use a validated instrument and correct but insufficient descriptive and variance analyses. Some information is missing, such as whether the homoscedasticity and normal distribution criteria necessary for ANOVA were met. Also, although the results are almost similar, as there are two independent samples (ASD and neurotypicals), the appropriate technique would have been the T-test for independent samples. Even non-parametric analysis (Mann-Whitney) would have been more appropriate since one group (Neurotypical) does not reach 30 individuals.  Homoscedasticity is impossible given the differences between the standard deviations in each group. In ANOVA, in fact, the pairwise post-hoc is a relevant result that is not applicable given that there are two groups.

It is also not enough to check the reliability of the instrument's dimensions, but also its validity through, for example, discriminant validity. A confirmatory factor analysis is indispensable. 

The results of the data analysis are too obvious. 

It is not helpful to offer a description that emphasises the obvious sensory differences between ASD and neurotypical. It would be more useful to offer a perspective on how to minimise these differences. This study would make a real contribution to scientific research if it had not remained merely diagnostic.

Reviewer 4 Report

Dear Authors,

I have just finished reviewing your manuscript and would like to highlight the following points: 

  1. Drawing conclusions from a self-administered questionnaire on such a small study cohort is quite dangerous, particularly given the complexities in the behavioural traits of children with ASD. Despite the control provided by the comparison with responses by neurotypical children, it would be appropriate to emphasise the fact that these results constitute a starting point from which to develop further hypotheses. 
  2. You discuss interventions in lines 230. I would rethink this paragraph to take into account my comments in point 1 above, and to discuss "observations", with a focus on qualitative, rather than on quantitative, especially when dealing with small cohort sizes. It is the rich data that you will obtain from interviews and practitioner observations that will help close the bias gap in a methodological stance purely based on quantitative methods. 
  3. As a suggestion, you may wish to include a future directions paragraph in which you may discuss a mixed-method approach as a natural direction to go forward. Data from semi-structured interviews of parents and teachers, processed by thematic analysis, triangulated with data from an autism behaviour inventory (completed by parents / caregivers, e.g. the inventory by Bangerter et al) as well as with your own questionnaire is likely to provide you with interesting insights.
  4. Please check for typos, e.g. "analyzes" in line 121. 

Hope you find my comments useful. 

Kind regards,

The reviewer 

References

Braun, V. & Clarke, V. (2006). Using thematic analysis in psychology. Qualitative Research in Psychology, 3, 77-101.DOI:10.1191/1478088706qp063oa

Braun, V. & Clarke, V. (2014). What can "thematic analysis" offer health and wellbeing researchers? International journal of qualitative studies on health and well-being, 9, 26152-26152.DOI:10.3402/qhw.v9.26152

Bangerter, A., Ness, S., Lewin, D., Aman, M. G., Esbensen, A. J., Goodwin, M. S., Dawson, G., Hendren, R., Leventhal, B., Shic, F., Opler, M., Ho, K. F. & Pandina, G. (2020). Clinical Validation of the Autism Behavior Inventory: Caregiver-Rated Assessment of Core and Associated Symptoms of Autism Spectrum Disorder. Journal of Autism and Developmental Disorders, 50, 2090-2101.DOI:10.1007/s10803-019-03965-7

Round 2

Reviewer 3 Report

Dear authors,

the great flaw of this work is the subject matter itself and the lack of novelty of the study. Therefore, I do not consider it pertinent to publish it in this context. 
